# Gas Imaging with Uncooled Thermal Imager

**DOI:** 10.3390/s24041327

**Published:** 2024-02-19

**Authors:** Mengjie Zhang, Guanghai Chen, Peng Lin, Daming Dong, Leizi Jiao

**Affiliations:** 1College of Electronic Engineering (College of Artificial Intelligence), South China Agricultural University, 486 Wushan Road, Guangzhou 510642, China; 20213162201@stu.scau.edu.cn (M.Z.); chenguanghai@stu.scau.edu.cn (G.C.); linpeng_hn@stu.scau.edu.cn (P.L.); 2Research Center of Intelligent Equipment, Beijing Academy of Agriculture and Forestry Sciences, 11 Shuguang Garden Middle Road, Beijing 100097, China; dongdm@nercita.org.cn; 3Key Laboratory of Agricultural Sensors, Ministry of Agriculture and Rural Affairs, Beijing 100097, China

**Keywords:** infrared imaging measurement, gas detection, uncooled thermal imager, development status

## Abstract

Gas imaging has become one of the research hotspots in the field of gas detection due to its significant advantages, such as high efficiency, large range, and dynamic visualization. It is widely used in industries such as natural gas transportation, chemical, and electric power industries. With the development of infrared detector technology, uncooled thermal imagers are undergoing a developmental stage of technological advancement and widespread application. This article introduces a gas imaging principle and radiation transfer model, focusing on passive imaging technology and active imaging technology. Combined with the actual analysis, the application scenarios using uncooled thermal imaging cameras for gas imaging measurement are analyzed. Finally, the limitations and challenges of the development of gas imaging technology are analyzed.

## 1. Introduction

Industrial gases play a crucial role in many modern economic activities and generate substantial value in energy, chemical, electricity, and agriculture sectors [1,2,3,4]. These gases constitute an indispensable part of both contemporary human life and industrial production. However, during the transportation, storage, and use of these gases, the occurrence of gas leaks is a common and often concerning issue. If these leaks are not detected and dealt with in time, under certain conditions, the leaks can easily cause explosions, fires, and other catastrophic events, posing a major safety hazard. In addition, the release of harmful gases such as methane and sulfur hexafluoride causes air pollution and climate change. People exposed to certain leaked gases can incur acute health problems and long-term chronic diseases [5,6].

Therefore, the world has been paying close attention to the research into gas imaging measurement technology and the development of related instruments. According to working principles, the most common gas detection sensors can be divided into semiconductors, electrochemical sensors, infrared absorption sensors, and other technologies [7,8,9]. However, these sensors are expensive, cannot perform detection in real time, and also risk exposing the detector to dangerous gases. In recent years, the evolution of gas imaging technology has led to the widespread adoption of optical gas imaging (OGI) technology as an efficient gas detection tool. OGI technology boasts several advantages over traditional gas detection methods, including the following:Real-time visualization capability [10];Rapid location of leaks;Short-time large-area scanning and imaging;Noncontact, remote monitoring capability [11];Continuous power monitoring [12].

The first infrared thermal imaging device was researched in 1947 and was usually used for night observation applications in the military. Thermal imagers are extensively applied in the electrical, aerospace, architectural, mineral material identification, and medical domains [13,14,15,16,17]. The human eye can only see a proportion of the electromagnetic spectrum, i.e., visible light, but infrared thermography can visualize infrared radiation that is invisible to the human eye. Infrared radiation is a form of electromagnetic waves, with wavelengths that are longer than those of visible light.

Any object with a temperature above absolute zero (i.e., T > 0 K) emits infrared radiation outwardly [18,19]. Initially, infrared thermography was not used widely among the general public because of the high manufacturing costs and high prices of the equipment required. One major obstacle to its wider use was that the operating temperature of the thermal imaging detector at the time of development was approximately 77 K, which required the use of expensive cooling devices. The cooling device effectively eliminates the thermally induced noise of the imaging device itself, thus making the detection area easier to identify. In the 1990s, thermal imagers were equipped with detectors based on barium strontium titanate technology, which allowed for their use at normal ambient temperatures. This simplified the thermal imager system’s design and reduced its cost to a level that was more accessible to the general public [20]. In recent years, the application of thermal imagers has become more and more widespread.

The detector is the core component of a thermal imager. Thermal imagers can be divided into cooled and uncooled thermal imagers, according to the conditions of use of their detectors. Of the two types, cooled thermal imagers offer advantages that include high sensitivity, high spatial resolution and a long-distance detection capability [21]. Cooled thermal imaging cameras typically have smaller pixel pitches [22]. However, their high costs, large dimensions, and short working times also increase the difficulty and uncertainty of their use [23]. With ongoing advances in detector manufacturing, the thermal imager has gradually overcome the constraints of low-temperature cooling during operation and large volume and has also compensated for the low sensitivity in ambient temperature. Uncooled thermal imagers can compensate for their lower resolution and sensitivity with further image processing [24]. Currently, uncooled thermal imagers can perform long-term continuous measurements and work with high precision under normal environmental temperature conditions. When compared with cooled thermal imagers, uncooled thermal imagers are more affordable and are accessible to both businesses and the general public. Uncooled thermal imagers are thus gradually expanding their share of the detection market and are expected to become the main technology for use in infrared thermal imaging measurement in the future [25].

Gases are selective emitters, and energy transfer occurs when infrared radiation passes through a gas cloud, causing the gas molecules to vibrate or rotate. Depending on its molecular composition, a specific gas will absorb infrared radiation at a distinctive wavelength. When a gas cloud absorbs radiation within a thermal imaging detector’s specific response range, the radiation detected by the imaging detector decreases, and this enables the formation of an image [26]. Thermal imaging systems generally consist of optical systems, infrared detectors, digital signal processing systems, and display systems. The optical system focuses the received infrared radiation on the photosensitive element of the infrared detector. Infrared detectors receive infrared radiation and convert it into electrical signals. The electrical signal is amplified and processed by a digital signal processing system. The display system visualizes the infrared radiation [27]. The working framework of the thermal imaging system is shown in Figure 1.

Here, we introduce a simplified radiation transmission model that divides the entire radiation transmission process into three layers, as shown in Figure 1. Layer 1 is the stage before the infrared radiation is absorbed by the gas cloud; layer 2 is the radiation before and when passing through the gas cloud (accounting for the absorption and radiation properties of the gas cloud); and layer 3 is the stage before the infrared radiation reaches the infrared lens of the thermal imager through the gas cloud [28]. The radiation difference in the thermal image originates from the upper and lower paths, as shown in Figure 2. The first path is the path by which the infrared radiation reaches the detector through the gas cloud, and it is called the gas path. The second path is the path by which the background radiation reaches the camera directly, and it is called the background path.

We assume a blackbody background system with no consideration of atmospheric scattering, a uniform temperature, a composition distribution in each gas layer, and a pure and transparent atmosphere. The infrared radiation $M_{off}(\lambda )$ in the background path to the optical lens can be expressed as follows:(1)$$M_{off}(\lambda )=\varepsilon_{B}M_{B}\left(\lambda_{3},T_{B}\right)$$
where λ is the wavelength, $M_{B}\left(\lambda_{3},T_{B}\right)$ represents the blackbody radiation emittance at temperature T, and $\varepsilon_{B}$ is the radiation emissivity of the background. The infrared radiation $M_{on}\left(\lambda \right)$ on the path of the gas can be expressed as follows:(2)$$M_{on}\left(\lambda \right)=\tau_{gas}\varepsilon_{B}M_{B}\left(\lambda ,T_{B}\right)+(1-\tau_{gas})M_{gas}(\lambda ,T_{g})$$
where $M_{gas}(\lambda ,T_{g})$ is the blackbody radiation emittance of the gas at the equivalent blackbody temperature $T_{g}$, and $\tau_{gas}$ is the spectral transmittance of the gas cloud. According to the Beer–Lambert Law, the spectral transmittance $\tau_{gas}$ of the cloud gas can be expressed as follows:(3)$$\tau_{gas}=e^{-\alpha (\lambda )\;c\;L}$$
where α(λ) is the infrared absorption coefficient of the gas, c is the gas cloud concentration, and L represents the absorption path length of the gas.

Therefore, the radiation difference between the gas path and the background path received by the thermal imager is as follows:(4)$$M_{on}-M_{off}=(1{-e}^{-\alpha (\lambda )\;c\;L})\;[M_{gas}\left(\lambda ,T_{gas}\right)-\varepsilon_{B}M_{B}\left(\lambda ,T_{B}\right)$$

The thermal imager detector will receive three kinds of radiation waves, which are generated by the target gas, the background, and the atmosphere. The atmosphere absorbs both visible light and infrared radiation. However, in the mid-wave infrared (3–5 μm) and long-wave infrared (8–14 μm) bands, the atmosphere has a lower absorbance and thus higher transmittance for the infrared radiation. Within these two wavelength ranges, the atmosphere has a relatively small effect on gas infrared imaging. This wavelength range is also known as the “atmospheric window”. The transmission spectrum of 1828.8 m of air is shown in Figure 3. Within this atmospheric window, specifically in the 7–16 μm wavelength range known as the “fingerprint region” of gases, absorption peaks exhibit high characteristic features [29]. These peaks can be used to discern subtle structural differences between different compounds. Consequently, gas infrared imaging measurements are generally conducted within this atmospheric window band.

In summary, the study of optical imaging measurements of gases with absorption peaks that occur in the atmospheric window using uncooled thermal imagers is of great significance and is a promising gas imaging approach. Most previous reviews of uncooled thermal imagers have tended to focus on construction [30], medicine, agriculture, and food [31] applications, and very few have summarized research on thermal imagers for gas imaging measurements. This review begins by describing the principle of gas infrared imaging, analyzes and summarizes the existing research on the application of uncooled thermal imagers to gas imaging measurements, and summarizes the current application scenarios of uncooled thermal imagers in gas imaging measurements in combination with some real-life applications.

## 2. Methods

This section mainly discusses two methods by which uncooled thermal imagers are used to perform gas imaging measurements: passive imaging technology and active imaging technology [10,19,32,33]. Passive thermal imaging technology relies solely on the infrared radiation emitted by gas clouds or background radiation. This technique has a wide detection range and can detect multiple types of gases, and is also able to detect gases at long distances [34]. In contrast, active thermal imaging is more suitable for scenarios where it is challenging to visualize the temperature contrast between the gas cloud and the background in the thermal image. In these cases, external heat sources are used to irradiate the gas cloud, enhancing the thermal contrast between the gas cloud and the background. Active imaging technology offers advantages such as a high signal-to-noise ratio (SNR), heightened sensitivity, and robust stability [35,36]. However, this technique comes with trade-offs in terms of increased system size and cost.

### 2.1. Passive Imaging Technique

Passive imaging technology relies on the radiation emitted by the gas cloud that is being tested or on the background radiation to perform imaging, and spectral band selection is achieved by using filters. Depending on the different types of gases to be detected, appropriate optical filters are added to the gas imaging system, and the characteristic infrared absorption peaks of the gases of interest are contained within the passbands of these filters [26]. Spectral selection is an essential technology that determines the types of gases that can be detected and the detection sensitivity directly [37]. The selection of available optical filters mainly includes bandpass filters and long-pass filters.

#### 2.1.1. Bandpass Filter

The use of bandpass filters for spectral selection has a minimal interference effect on the imaging process. By switching filters through a filter wheel, it is possible to detect multiple gases and estimate their concentration ranges. However, one drawback of this approach is that the data processing becomes more complex, and the technical challenges are also heightened.

In cases where only one type of gas is detected, and particularly in the case where only one or a few narrow infrared absorption peaks are present, narrowband filters are often used to perform spectral selection. This is largely because of their advantages, which include simple structures and high cost-effectiveness [38]. Using narrowband filters that match the strong absorption bands of the target gases can increase the percentage of the radiation that is absorbed by the detector as a result of gas absorption [39,40]. The proportion of the yellow region shown in Figure 4a is significantly smaller than the corresponding region in Figure 4b. Common gases including NH_3_, SO_2_, NO, CH_4_, and SF_6_ have narrow absorption peaks in the infrared band and are thus suitable for spectral band selection using a single filter.

In 2000, Jonas Sandsten et al. [41] first developed the gas passive imaging method using thermal imagers in combination with narrowband filters and gas-cell-related techniques, thus enabling gas recognition. They used windpipes filled with ammonia, ethylene, and methane, which they hung from trucks to simulate gas leaks. They demonstrated gas separation techniques with overlapping spectra, presented real-time images of ammonia, ethylene, and methane leaks, and showed the separation of two different gases as they escaped simultaneously. Several optical imaging systems have since been tailored for individual gases by selecting a narrowband filter that corresponds to a specific wavelength. Notably, the GasFindIR series thermal imager from FLIR in the USA has been instrumental in these endeavors and has demonstrated excellent capabilities for the detection of gases including methane, propane, and butane [42,43].

Sulfur dioxide is a major component of the volcanic plumes that are emitted from volcanoes and from some industrial sites. Prata et al. [44] measured the absorption characteristics of sulfur dioxide at 8.6 μm using a spectral bandpass filter with a wavelength of 500 nm to monitor the gas. This approach was used to avoid excessive interference from water vapor, which can be caused by filters with excessively wide bandwidths. In 2014, Prata et al. [45] used a spectral bandpass filter ranging from 8.2 μm to 9.2 μm to detect sulfur dioxide absorption peaks at 8.3 μm in order to avoid the influence of water vapor in the atmosphere on imaging results. This represented the first observation of a volcano using a ground-based uncooled thermal imager.

Gas imaging measurements that are performed using a single narrowband filter are often less cost-effective. In practical applications, researchers generally use a filter wheel in combination with a wide-band detector to achieve a wider variety of gas imaging measurements [46]. In this approach, a specific filter will cover the infrared absorption peak of a particular gas, while the detector covers the overall infrared characteristic absorption peaks of multiple gases. Jin et al. [40] optimized the production of an uncooled infrared focal plane array (UFPA) that can cover the 3–12 μm band. They used a transmission optical lens to increase the space between the optical lens and the detector for installing an electric filter wheel [37]. Their overall system structure is shown in Figure 5. Different sub-band filters with center wavelengths (CWLs) of 3 μm, 3.5 μm, 4.3 μm, 4.7 μm, 10.6 μm, and 10.9 μm can be used in this system to detect methane, formaldehyde, carbon dioxide, carbon monoxide, ammonia, and ethylene, respectively. The filter wheel is driven using a stepper motor controlled by a program module. The 14-bit digital infrared image obtained is improved using infrared image non-uniformity correction (NUC) technology and digital detail enhancement (DDE) technology to enhance the gas plume, which can then be observed easily by the human eye [47,48].

Bertin’s Second Sight gas imager was the first commercially available uncooled thermal imager designed for the routine detection of alkane gases in a diverse range of environments. The Second Sight TC system is equipped with an electric filter wheel that can hold up to six filters. By loading filters that target narrowband absorption peaks of different gases, it is then possible to detect up to six types of gases simultaneously. This system is commonly used for detecting industrial and combustible gases [49,50]. Additionally, a broadband detector equipped with multiple narrowband filters can also distinguish between two gas species with absorption peaks that are close together in the detection range. For example, SF_6_ has one absorption peak at 10.55 μm, while NH_3_ has two absorption peaks at 10.35 μm and 10.73 μm, which are all very close together. The gas type cannot be determined using only two filters in this case and must therefore be distinguished by using multiple filters for all absorption peaks of the gas. This method of gas imaging using a filter wheel with multiple filters in combination with a broadband detector is now mature and is used widely in industrial production.

Although narrowband filters can provide thermal images with higher thermal contrast when compared with detectors without such spectral selection technology, they also restrict the radiation that actually reaches the detector, thereby reducing the thermal imager’s sensitivity, reducing its SNR, and increasing the thermal imager’s noise equivalent temperature difference (NETD) [51].

During actual imaging measurements, if the filter bandwidth is too narrow, it can cause the signal to be drowned in detector noise. The use of broadband filters can ensure that sufficient radiation reaches the detector, thereby improving the SNR of the resulting infrared images. Through the careful selection of the filter band parameters and the gas absorption lines, the visibility of gases within the selected filter image can thus be enhanced.

In our previous research [52], we used a gas imaging system that consisted of a standard blackbody, sample containers, infrared absorption filters that were specially designed for ethanol, and thermal imagers. The passbands of the filters were from 9μm to 10μm. The accumulated gray value (AGV) and the actual imaging area (AIA) were used as semi-quantitative analysis metrics to allow slightly rotted grapes to be separated quickly from severely rotted grapes. In another study by Xu et al. [53], Spectrogon’s Bandpass Interference Filters (BBP-3000-5000) were used in conjunction with a broadband uncooled IRFPA to perform methane gas detection. Two filter types were selected to target the mid-infrared and long-wave infrared absorption peaks for methane, with spectral passbands of 3–5 μm and 6.5–8.5 μm, respectively. This configuration achieved an imaging detection effect of 0.2 L/min.

When using broadband filters for infrared imaging, the main factor influencing the imaging performance is the radiation fluctuation caused by the temperature variation of the filter itself, but this can be reduced through a filtering radiation correction process using environmental blackbody radiation subtraction [54]. Figure 6 shows a comparison of the filters before and after radiation correction. The non-uniformity in the rectangular area of filter 4 and filter 5 have been corrected to become clear and uniform.

#### 2.1.2. Long-Pass Filter

Long-pass filters, which are often used in combination with multiple filters, offer the advantage of the ability to detect a broad range of gases at lower cost. In 1991, the French company Bertin proposed a gas leak infrared imaging detection technology based on differential spectral filtering [55]. Differential spectral imaging technology offers advantages that include a high SNR, a wide spectral range, and high-cost performance. The technique uses two filter types: a reference filter and an active filter. The passband of the reference filter is located outside the target gas absorption range, and the passband of the active filter contains the absorption peaks of the target gas. As illustrated in Figure 7, their technique can extract a target gas via differencing of the IR images obtained using the two neighboring filters, or it can differentiate between different gases with absorption peaks that are close together. Differential infrared imaging detection can reduce the influence of radiation and reflection from the filter itself effectively and can thus improve the sensitivity of uncooled infrared imaging systems.

The dual-band spectral differential (DB–SD) infrared gas imaging system commonly uses two long-wave pass filters. The strong absorption peak of the target gas is located within the spectral region where the two filters do not overlap. The subtraction of the corresponding thermal images between the two filters thus yields a differential image [56]. Wideband long-wavelength filters ensure high irradiance, and they are imaged using two filters with different wavelengths. The differential image obtained via this differential operation is equivalent to direct imaging while using a narrowband filter, but this approach can acquire more of the incident radiation and enhance the contrast between the gas and the background. This improves the gas detection capability of the DB–SD system effectively. Figure 8 presents a schematic diagram of the DB–SD gas enhancement method.

Ma et al. [57] used multiple wideband long-pass differential filters to create an infrared imaging device. The spectral response range of their device is from 7.5 μm to 13.5 μm. By using four long-pass filters with cutoff wavelengths of 9.6 μm, 10.15 μm, 11.3 μm, and 11.765 μm, they were able to calculate five equivalent narrow bands, i.e., 7.5–9.6 μm, 9.6–10.15 μm, 10.15–11.3 μm, 11.3–11.765 μm, and 11.765–13.5 μm, thus enabling the identification of C_2_H_2_ and SF_6_, which was achieved successfully. Li et al. [58] constructed a multispectral infrared imaging gas detection system using a combination of six long-pass and short-pass filters. Their system performs real-time processing of the collected infrared images and can detect NH_3_, SF_6_, CH_4_, and SO_2_ gases in real time.

When compared with imaging systems based on narrowband filters or on narrowband detectors, differential filter infrared imaging technology requires a longer response time because of the requirement to switch between different filters. Although the first two imaging methods can detect gas leaks and pinpoint the leakage position, they face challenges when attempting to identify specific gases. In contrast, differential imaging can approximate the type of gas and also determine the leakage position simultaneously [53]. The differential operation can not only enhance the image contrast between the gas and the background, but also eliminates the fixed pattern noise from the infrared images. However, differential processing inevitably accumulates random noise, which then leads to deterioration in the image quality. Furthermore, for differential images with only small amounts of gas leakage, there are further issues that include weak contrast and unclear boundaries. The above shortcomings can be improved by infrared image processing methods to improve detection accuracy. Common processing algorithms include non-uniform correction, digital enhancement, and denoising. These algorithms can improve the quality of infrared images and improve detection accuracy [23].

### 2.2. Active Imaging Technique

Most the research discussed above was conducted in laboratory environment conditions or assumed a blackbody background. That means that there is a thermal contrast between the gas-free cloud region and the gas cloud region, but if the temperature difference between the gas and the background is not large enough to ensure that the temperature difference is less than the NETD of the uncooled thermal imager, the thermal imager will then be unable to image the gas cloud to be measured accurately. In this case, it becomes necessary to illuminate the gas using an external light source to increase the temperature difference between the gas and the background [28], as shown in Figure 9. The light source used can be sunlight or it can be an artificial light source, e.g., low-cost broadband tungsten halogen lamps, narrowband lasers, or light-emitting diodes. In this case, the emitted light should match the energy gap of the gas to be measured [59,60].

Active imaging technology uses a laser as the light source to illuminate the gas to be measured, and the detector then receives the reflected light for imaging. The use of such a tunable narrowband laser as the primary illumination source is independent of any background radiation, and thus provides lower detection limits and better spectral resolution. Nutt et al. [61] constructed a portable methane detection imager using InGaAs laser diodes and InGaAs focal plane sensors that weighed 3.2 kg. This imager can be installed in a kitchen ceiling to monitor leakage from natural gas pipelines. With a detection range of up to 3 m, this imager has a lower detection flow limit of 0.05 L/min. Li et al. [62] detected leaked methane gas using Tunable Diode Laser Absorption Spectroscopy (TDLAS) technology. A schematic diagram of their active laser imaging system is shown in Figure 10. They used a semiconductor laser with a central wavelength of 1.653 μm to irradiate methane gas at a distance of 10 m. A leakage rate of 1 L/min was detected, and the imaging effect is quite evident.

Active imaging using an external light source is more suitable for gas detection in small areas than passive imaging techniques based on background radiation. Long-distance irradiation can cause uneven illumination in the area that is being tested, thereby degrading the imaging effect [63]. Achieving uniform laser illumination by using a small aperture, a large field of view, and long-distance irradiation is another hot research topic.

Active imaging techniques based on uncooled detectors are generally suitable for long-distance and large-area gas imaging. However, when high precision and high sensitivity are required in the imaging measurement system, researchers tend to use cooled detectors in combination with external light sources to perform imaging measurements of the target gas [64].

## 3. Application Scenario

Infrared imaging technology has characteristics that include noncontact measurement, long-distance use, convenient operation, and high sensitivity. This approach is used widely in the visualization of volatile organic compound (VOC) gases, the visualization of oil and gas, along with other gases in industry, the detection of electronic equipment, and the identification of leaks in petrochemical pipelines [65]. This section summarizes and analyzes various application scenarios of gas imaging measurements using uncooled thermal imagers, as listed in Table 1.

### 3.1. Industrial Source Gases

Common industrial source gases include CO_2_, CH_4_, NO_x_, SO_2_, VOCs, etc. By combining thermal imaging monitoring technologies, the industrial sector can prevent and manage the leakage and emission of harmful gases more effectively, thereby improving both production safety and environmental protection levels. When detecting toxic gases, there is a potential poisoning risk to the detector personnel. The installation of thermal imagers at locations in the plant that are prone to leakages would thus ensure the personal safety of the inspectors, in addition to enabling the detection of leaks in the plant [71].

The primary locations for methane leakage are in natural gas transportation facilities and at petrochemical plants. Leakage can occur at any stage from production, processing, and transportation to usage, which results in millions of tons of methane being leaked annually [72]. Thermal imagers are used widely in oil and gas refineries, petrochemical plants, and offshore platforms to monitor methane leakage [73]. The working and operating environments of oil and gas refineries can be harsh and challenging. For example, refineries in the Middle East operate in temperatures of up to 323.15 K, often with sandstorms [74]. Most thermal imagers can operate in ambient temperatures from 258.15 K to 333.15 K and are resistant to the effects of water, dust, and vibration [75]. Industrial vessel leakage detection in refineries has strict requirements for reducing downtime. Thermal imagers for large interview scanning detection can effectively reduce downtime [76], thus reducing economic losses and environmental pollution. In addition, thermal imagers are also used to monitor methane in landfills [66]. In a landfill, various types of organic waste can easily be decomposed by bacteria to produce methane in an anaerobic environment. The explosion limit for methane in the air is 5%. Therefore, explosions are highly likely to occur in landfill sites. The accuracy and efficacy of uncooled thermal imagers can be significantly influenced by environmental factors, such as ambient temperature fluctuations, humidity, and the presence of other heat sources. These factors can create false positives or mask the thermal signature of gas leaks, particularly in outdoor or complex industrial settings.

The main source of sulfur dioxide in the atmosphere is from the industrial processing of sulfur-containing resources, including the use of sulfur-containing fossil fuels in thermal power plants. Sulfur dioxide can be readily combined with other substances to form harmful chemicals, including sulfuric acid, acid rain, and sulfate particles. These chemicals pose risks to the environment, public health, and the climate in general [77]. In addition, volcanic eruptions also produce large quantities of sulfur dioxide gas. The measurement of gas emissions from volcanic plumes is important for eruption prediction and surveying applications. As outlined in Section 3.2, significant numbers of researchers use thermal imagers to detect and measure the gases produced by volcanic eruptions, and the levels of carbon dioxide and sulfur dioxide emissions are important parameters in the study of volcanoes.

### 3.2. Greenhouse Gas

Greenhouse gases (GHGs) refer to the natural and anthropogenic gaseous components in the atmosphere that absorb and re-emit infrared radiation. The GHGs specified in the Kyoto Protocol include CO_2_, CH_4_, N_2_O, hydrofluorocarbons (HFCs), perfluorocarbons (PFCs), and SF_6_ [78]. According to the World Resources Institute, the current composition of the global emissions of GHGs is as shown in Figure 11. Increases in the presence of GHGs can lead to glacier retreat, rising sea levels, northward shifts of climate zones, and more frequent occurrences of the El Niño phenomenon, and will consequently trigger a series of ecological problems.

Natural CO_2_ mainly comes from the respiration of plants and animals, volcanic eruptions, and the decomposition and decay of organic matter. There is a significant increase in the amount of CO_2_ released by volcanoes before eruptions and a significant increase in the amount of SO_2_ released after the eruptions. Therefore, these two gaseous phenomena are often used as parameters in the study of volcano characteristics and also in the early detection of volcanic eruptions [79]. Researchers often use thermal imagers to monitor changes in gas species and their concentrations before and after volcanic eruptions [80,81]. It is also important to monitor the effects of volcanic eruptions on the local and global climate systems. Carbon dioxide from anthropogenic activities is mainly the result of the burning of fossil fuels and industrial production. Geological carbon sequestration offers the potential to reduce the release of GHGs from the combustion of materials into the atmosphere [82,83,84]. Thermal imagers can detect whether sequestered CO_2_ continues to remain in the ground without leaking into the atmosphere [85].

Sulfur hexafluoride, which has excellent electrical insulation properties and other advantages, is used extensively in high-voltage equipment, insulation equipment, and circuit breakers [86]. SF_6_ gas leakage in substations is one of the most common problems during the operation of gas-insulated high-voltage electrical equipment. When SF_6_ leaks into the air, it produces 23,000 times the greenhouse effect of carbon dioxide in an equivalent volume [87,88]. The traditional detection methods include soapy water detection and wrapping methods [89]. The traditional detection methods cannot achieve large-area detection and must be used during a power outage, which greatly affects the detection of SF_6_ [86]. Cambridge University researchers have designed a fully autonomous robot system, named Longsword, which utilizes a thermal imager and robots to conduct inspections in indoor substations. At present, more than 200 Longsword robots are servicing approximately 160 distribution rooms [90].

Vehicle exhaust gases represent another significant source of GHGs. The thermal signature of these exhaust gases serves as a valuable tool for the visualization and analysis of vehicle exhaust emissions. Jain et al. [91] used a thermal imager in conjunction with a gas analyzer to monitor the exhaust emissions of trucks and buses in various states.

Whether monitoring industrial source gases or greenhouse gases, intelligent gas detection algorithms play an indispensable role. The infrared images captured by thermal imaging systems require processing to derive vital data such as the location, type, and concentration of gas clouds. The typical workflow of gas detection algorithms involves several key steps: (1) preprocessing, which includes enhancing and denoising the image; (2) extracting the foreground to isolate the target area; and (3) conducting feature screening for accurate detection.

The main reason for preprocessing is that it is not easy to obtain the target gas cloud area in real scenes, and noise will interfere with detection. Preprocessing can enhance infrared image details and reduce noise. In order to solve the problems of low signal-to-noise ratio and weak imaging of toxic gas infrared images, Liang et al. [92] proposed a real-time gas image enhancement algorithm based on a guided filter and simplified kernel gain mask to improve the visual effect of gas infrared images.

Foreground extraction in infrared imaging involves segmenting the background and foreground to isolate the gas cloud regions within the foreground. This approach often employs a method of ‘motion target extraction followed by judgment.’ Wang et al. [93] developed the first deep learning-based infrared gas detection algorithm model, GasNet, specifically for detecting methane gas leaks. This model utilizes background modeling to extract foreground motion targets, then applies an improved convolutional neural network model to determine whether these motion targets are gas clouds. Weng et al. [94] implemented frame differencing on infrared videos to identify moving target areas. Subsequently, they extracted SIFT features and employed the Support Vector Machine (SVM) algorithm to analyze these features, thereby detecting gas cloud targets.

In the realm of gas cloud imaging, commonly analyzed features include motion, shape, and texture characteristics. The Histogram of Oriented Gradient (HOG) is a method used for computing the statistical values of local image gradient orientations, which assists in describing the local texture features of an image. Hong et al. [95] employed a combination of HOG and Support Vector Machine (SVM) algorithms, utilizing the shape and texture features of gas clouds for the differentiation and localization of Volatile Organic Compound (VOC) gas clouds. This approach significantly reduced the rate of false positives. When supported by appropriate hardware and thorough training, gas detection methods based on deep learning exhibit enhanced speed and greater generalization capabilities. These methods eliminate the need for manual feature design, simplifying preprocessing operations and thereby presenting a promising future for the development of intelligent gas detection techniques.

## 4. Conclusions

Over the last decade, infrared thermal imaging technology has evolved from its origins in military applications to becoming a routine inspection tool, catering to a myriad of engineering needs. This review commences with an examination of the principles underlying optical gas imaging, subsequently delving into a detailed comparative analysis of the merits and demerits associated with both cooled and uncooled thermal imagers. The article delineates the methodologies of active gas imaging, which relies on external light sources, and passive gas imaging, which operates independently of such sources. It also illustrates the preferred contexts for deploying uncooled thermal imagers in gas detection.

Synthesizing existing scholarly findings, it is evident that infrared thermal imaging technology continues to advance in accuracy, intelligence, automation, and portability. Nonetheless, optical gas imaging technology currently encounters several limitations and challenges:Infrared Gas Intelligent Detection Algorithms: There is a necessity for enhancement in intelligent detection capabilities. Presently, the identification of gas targets predominantly depends on the expertise of professionals. Uncooled thermal imager cameras offer a relatively precise quantification of gas clouds, encompassing measurements of column density, gas path concentration, and leak rate. However, the real-time quantification accuracy for leaked gases requires further improvement. A significant obstacle in the amalgamation of artificial intelligence with optical gas imaging detection is the absence of high-quality, objective public datasets of gas infrared images for training models, rendering subjective datasets inadequate for an objective evaluation of algorithmic performance.Infrared Detectors for Optical Gas Imaging: Most existing optical imaging systems employ general-purpose detectors, supplemented with specialized filters and image processing technologies for gas imaging measurements. The efficacy of thermal imaging systems hinges on the application of these professional filters and advanced thermal image processing technologies. Importantly, the development of highly sensitive uncooled infrared detectors, specifically designed for gas imaging measurements, stands as a critical area for advancement and poses considerable challenges.Inherent Limitations of Thermal Imaging Cameras: Despite their utility, thermal imaging cameras exhibit inherent limitations, such as low contrast, limited resolution, and indistinct boundaries between targets and backgrounds in the captured thermal images. Furthermore, various noise sources can impair image quality. Addressing these challenges necessitates the preprocessing of the captured thermal images, utilizing specialized algorithms for non-uniformity correction, enhancement, noise reduction, background modeling, and feature extraction.Environmental Impact on Thermal Imaging Performance: The operational environment of thermal imaging cameras significantly influences their performance. In extreme heat, lens and window materials may deform or discolor, altering infrared radiation transmission and reflection and potentially leading to measurement inaccuracies. Camera performance can also be affected by thermal effects like thermal drift in high-temperature settings. In high-humidity environments, increased water vapor density can enhance infrared radiation absorption and attenuate radiation from target objects, thereby hampering effective detection, particularly on foggy or rainy days. While wind speed does not directly impact camera performance, it influences gas cloud behavior, accelerating gas diffusion and increasing the Minimum Detectable Leak Rate (MDLR).Miniaturization and Integration: Advancements in technology are driving the trend towards the miniaturization and integration of thermal imaging equipment. Future developments might see these cameras becoming more compact and seamlessly integrated with other surveillance systems, broadening their applicability across diverse applications.

## Figures and Tables

**Figure 1 sensors-24-01327-f001:**
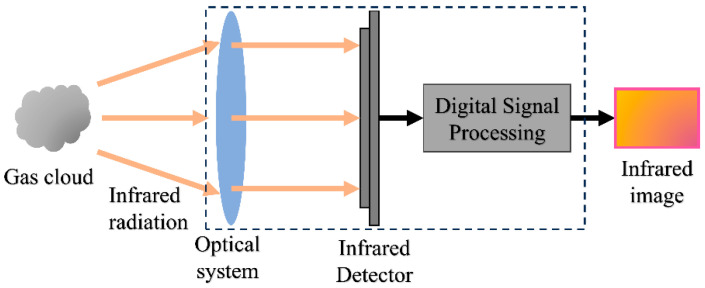
Thermal imaging system composition and working principle.

**Figure 2 sensors-24-01327-f002:**
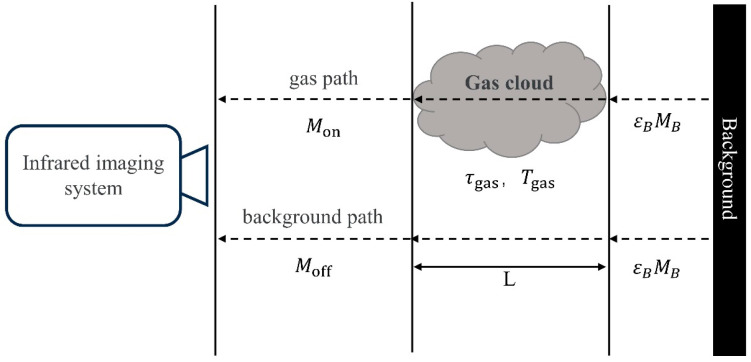
Diagram of three-layer radiation transmission model.

**Figure 3 sensors-24-01327-f003:**
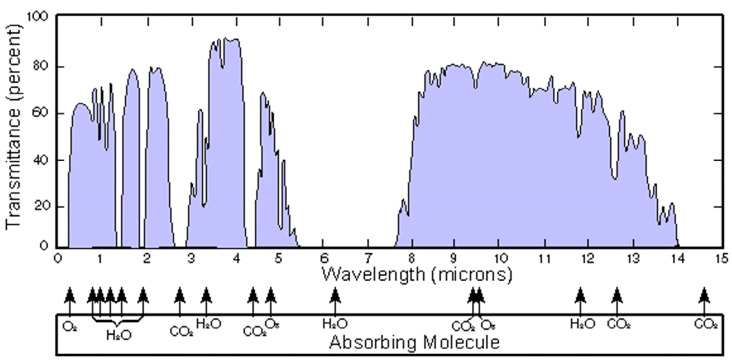
Atmospheric transmission spectrum at 1828.8 m (image from [13]).

**Figure 4 sensors-24-01327-f004:**
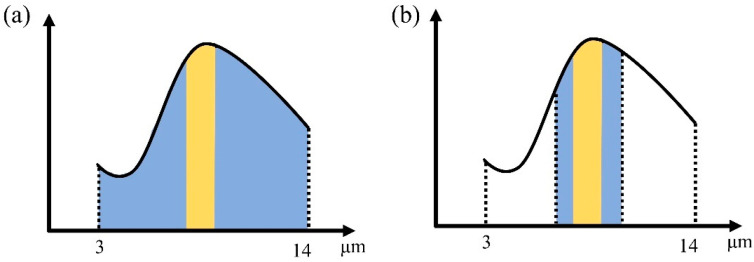
The Influence of narrowband filters on the response band of specific wavelength radiation. The blue colour indicates the transmission band of the filter and the yellow colour indicates the gas absorption band. (**a**) Broadband mode. (**b**) Narrowband mode.

**Figure 5 sensors-24-01327-f005:**
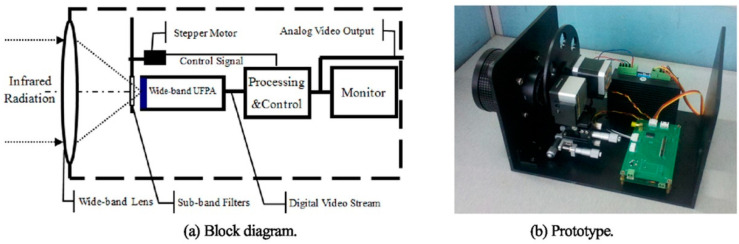
The wide-band gas leak IR imaging detection system.

**Figure 6 sensors-24-01327-f006:**
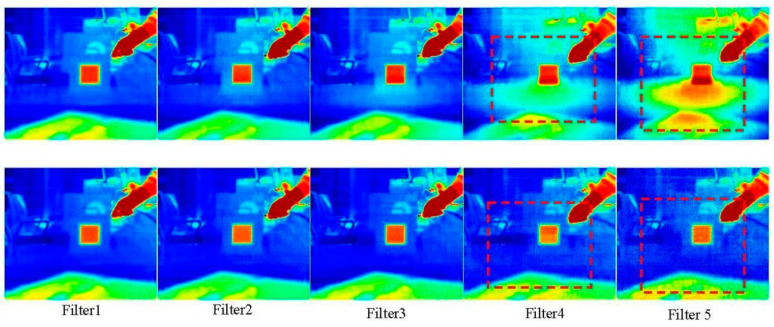
The top and bottom rows show the thermal images before and after calibration, respectively.

**Figure 7 sensors-24-01327-f007:**
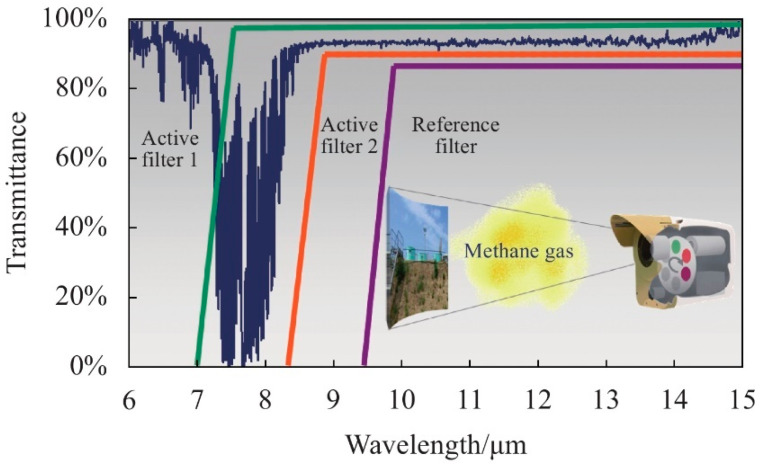
Schematic diagram of the application of active and reference filters.

**Figure 8 sensors-24-01327-f008:**
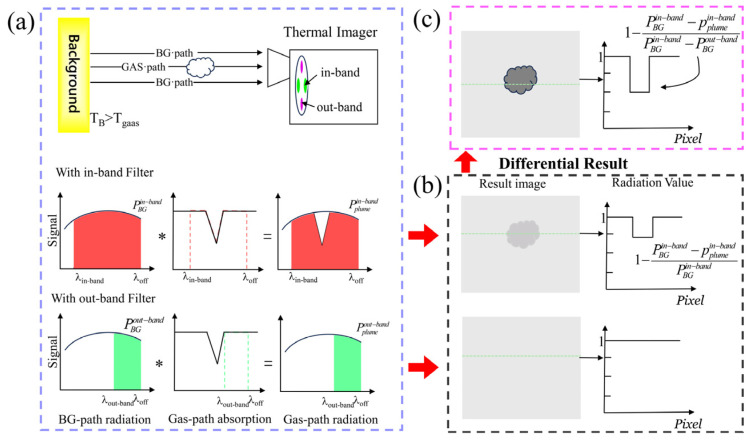
Schematic diagram of the DB–SD gas enhancement method. (**a**) The top row shows the radiation transfer for DB–SD imaging, and the bottom two rows are the in-band and out-band gas path and the BG path filtered by the IR radiation, respectively.∗ represents the operation of the filter on incident light.(**b**) the in-band and out-band images and their corresponding normalized grey values; and (**c**) the DB–SD image obtained by subtracting between the in-band and out-band images.

**Figure 9 sensors-24-01327-f009:**
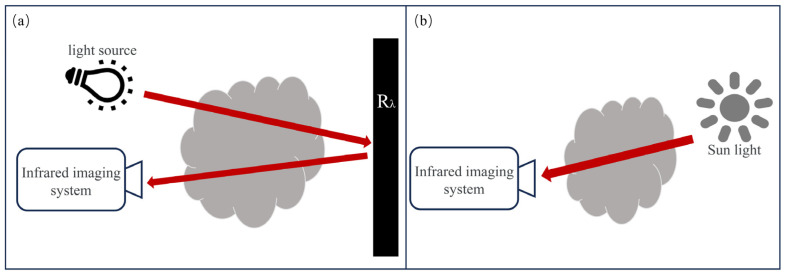
Schematic diagram of active imaging technology. (**a**) Artificial light. (**b**) Natural light source.

**Figure 10 sensors-24-01327-f010:**
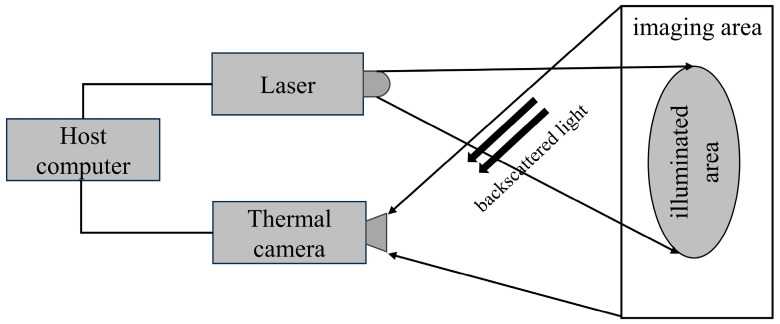
Schematic of active laser imaging system.

**Figure 11 sensors-24-01327-f011:**
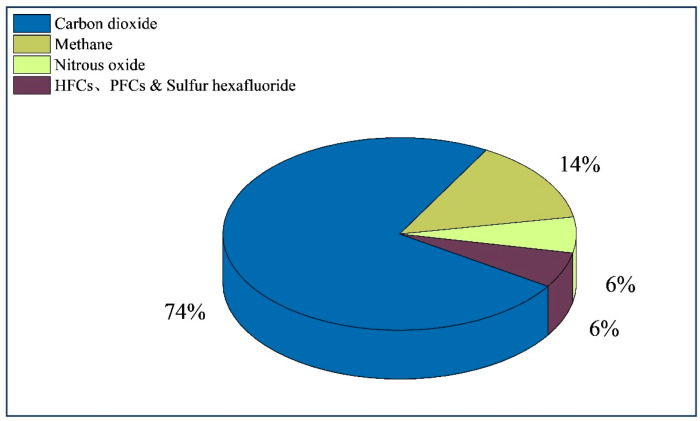
Share of greenhouse gas emissions in 2022.

**Table 1 sensors-24-01327-t001:** Summary of application scenarios of uncooled thermal imagers in gas imaging measurements.

Target Gas	Absorption Peak Band	Detection Limit/Concentration	Detection Distances	Application Scenario	Detection Reasons	Ref.
Methane	6.6 μm–8.6 μm	5 mL/min	300 m	Petrochemistry	Greenhouse gases, explosion-proof, reduced product loss	[12,53,60,61,66,67]
				Natural gas transportation		
				Kitchen		
				Landfill		
Sulfur hexafluoride	10.56 μm	≤0.1 L/min	200 m	Substations	greenhouse gas	[37,54]
Carbon dioxide	9.5 μm, 10.5 μm	5 L/min	600 m	Hospital guardianship	Asphyxiation, greenhouse gases	[8,59,68]
				Volcanic research		
Ethanol	9–10 μm	300 ppm·m	10 m	Fruit decay monitoring	Food quality testing	[52]
Sulfur dioxide	7.3 μm	3 ppm·m	600 m	Chemical plant	acid rain	[45,49]
				Volcano monitoring		
Ethylene	10.5 μm	10–100 L/min	5 m	Fruit Transportation	gaseous phytohormone	[37,41,57]
Ammonia	5.6 µm, 10 µm	200 ppm·m	20 m	Chemical manufacturing	Toxic to the human body	[58,69,70]
				Refrigerant		
Carbon tetrachloride	7.8 μm	-	20 m	Low-temperature refrigeration	Toxic to the human body	[9]

## Data Availability

Data are contained within the article.

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
