# Peer review of "Gas Imaging with Uncooled Thermal Imager"

_sensors, 2024, doi:10.3390/s24041327_

Round 1

Reviewer 1 Report

Comments and Suggestions for Authors

One thing that can be improved is units.

Sometimes wavelength is used and sometimes wavenumber is used.

Sometimes the wrong unit Feet is used.

The authors should check consistency in using SI units.

For Table 1 there is something wrong with the layout.

Table 1:

Title is unreadable

Target gas column, use name or formula not a mixture

The column Filter can be removed

The column Detection limit is very confusing.

The Longsword robot system has a confusing Figure 11. It is badly described.

Reviewer 2 Report

Comments and Suggestions for Authors

According to the paper's content and descriptions, several recommendations are proposed for enhancing the paper:

1. Additional details for the model description:

   - In the theoretical section, the article should delve deeper into explaining the operational principles of optical gas imaging technology. This involves providing a more elaborate explanation of sensor principles and the thermal radiation transfer model. Such an approach would empower readers to gain a profound understanding of the physical principles underpinning thermal imaging instruments, ultimately improving the paper's overall clarity.

   - Beyond the theoretical model for gas imaging, the paper should encompass the principles of algorithms related to gas imaging technology. This broader coverage will afford readers a more holistic comprehension of the model, facilitating a better grasp of research outcomes and promoting reproducibility.

2. Incorporation of a comparative analysis of gas imaging systems:

   - A meticulous comparative analysis of diverse gas imaging systems is needed, encompassing the pros and cons of both cooled and uncooled thermal imagers. However, the paper lacks additional technical details and performance metrics like spatial resolution and detection sensitivity. It is advisable to integrate more supporting evidence and data from other papers to fortify the article's claims.

   - While the article touches upon the working principles of optical gas imaging technology, it fails to furnish detailed information on various detection methods and techniques. The inclusion of more nuanced information would enable readers to better grasp and apply these methods.

   - There is an necessary to expand the paper's discussion on the application scenarios of uncooled thermal imagers in gas imaging measurements, including their application across diverse gas types and fields.

   - The addition of real-world application cases and experimental results would serve to validate the feasibility and efficacy of gas imaging technology.

3. Necessary modifications in the conclusion section:

   - Despite the extensive discussion on the application scenarios and advantages of non-refrigerated thermal imagers in natural gas imaging measurements, the paper lacks an in-depth exploration of potential limitations or challenges. Furthermore, the conclusion section overlooks providing a performance evaluation or case studies involving uncooled thermal imagers within specific application domains.

4. Refinement in the principles, algorithms, and outlook sections:

   - In the algorithm section, the paper can offer more insights into the integration of thermal imagers with artificial intelligence to enhance gas detection accuracy and automation. Specifically, in the technical discussion section, additional information about the challenges faced by gas imaging detection, the current status, methods, and approaches of gas imaging visual detection algorithms could be incorporated.

- In the outlook section, a more profound exploration of how the development of artificial intelligence can make optical gas imaging systems more intelligent and automated would be beneficial. This could also highlight the potential applications of existing machine vision algorithms in gas detection scenarios.

Comments on the Quality of English Language

1" 3.1. Subsection": The identifier " 3.1. Subsection" lacks specific content, making it challenging to evaluate its accuracy. Ensure that section identifiers provide clear indications of the content they represent, helping readers and reviewers understand the context.

2"The paper compares the advantages and disadvantages of cooled and uncooled thermal imagers, highlighting the development of uncooled thermal imagers as an important trend": The use of the word "compares" suggests a direct and detailed comparison, which may not be accurately reflected in the content. It would be more precise to state that the paper discusses the advantages and disadvantages of both cooled and uncooled thermal imagers and emphasizes the significant trend in the development of uncooled thermal imagers.

3"The introduction section emphasizes the importance of gas detection due to the risks posed by gas leaks during transportation, storage, and usage": While generally accurate, the phrase "risks posed by gas leaks" is quite broad. To enhance accuracy, specify the particular risks associated with gas leaks, such as safety hazards, environmental impact, or health risks. Providing more specific information will strengthen the introduction and provide a clearer understanding of the context for readers.

Reviewer 3 Report

Comments and Suggestions for Authors

The work titled "Gas Imaging with Uncooled Thermal Imager" discusses the development and application of gas imaging technology using uncooled thermal imagers. The paper compares the advantages and disadvantages of cooled and uncooled thermal imagers. It introduces the detection principle of gas imaging technology and provides examples of representative gas imaging measurement systems. It also analyzes the application scenarios of uncooled thermal imagers in gas imaging measurement, emphasizing their efficiency, large range, and dynamic visualization capability. Overall, the paper emphasizes the potential of uncooled thermal imagers in gas imaging measurement applications. Generally, the manuscript can easily be understood, however, the paper could be improved by discussing more details. There are several remarks needed to be taken into account before acceptance:

1. Can you provide more details on the data analysis techniques used in gas imaging measurements? How are the thermal images processed and analyzed to identify and quantify the presence of gases?

2. Are there any specific challenges or limitations associated with the use of thermal imagers for gas imaging? How do environmental factors, such as temperature and humidity, affect the performance of these imagers?

3. What are the key factors to consider when selecting optical filters for gas imaging measurements? How do these filters impact the detection sensitivity and accuracy?

4.  It would be helpful to discuss the the traditional method with novel infrared imaging techniques. Some works can be referred in the background or introduction part. For example, Crystal Growth & Design, 2023, 23(11),7992-8008, DOI: 10.1021/acs.cgd.3c00780.

5. Including a section on future directions and potential research areas in the field of thermal imaging for gas detection would be valuable for researchers and industry professionals.

Comments on the Quality of English Language

Good

Round 2

Reviewer 2 Report

Comments and Suggestions for Authors

 1. The abstract section states, "This paper first discusses the advantages and disadvantages of cooled and uncooled thermal imagers," yet there is no detailed explanation of uncooled thermal imagers within the main body of the text. 

2. Detection distance in gas remote sensing serves as a vital metric with significant practical implications. It is advisable to include statistical data on detection distances in Table 1 for comprehensive reference. 

3. In the third section, "3. Application scenario," the text mentions the detection of gases with features in mid-wave and long-wave infrared. The primary strength of Optical Gas Imaging (OGI) lies in its real-time gas imaging capabilities. To align with the conclusion, it is suggested to incorporate content on "Infrared Gas Intelligent Detection Algorithms."

Comments on the Quality of English Language

The reference list contains Chinese characters, and the formatting is not consistent.
